# Denoised Smoothing:
# A Provable Defense for Pretrained Classifiers

**Hadi Salman**
hasalman@microsoft.com
Microsoft Research

**Mingjie Sun**
mingjies@cs.cmu.edu
CMU

**Greg Yang**
gragyang@microsoft.com
Microsoft Research

**Ashish Kapoor**
akapoor@microsoft.com
Microsoft Research

**J. Zico Kolter**
zkolter@cs.cmu.edu
CMU

## Abstract

We present a method for provably defending any pretrained image classifier against $\ell_p$ adversarial attacks. This method, for instance, allows public vision API providers and users to seamlessly convert pretrained non-robust classification services into provably robust ones. By prepending a custom-trained denoiser to any off-the-shelf image classifier and using *randomized smoothing*, we effectively create a new classifier that is guaranteed to be $\ell_p$-robust to adversarial examples, without modifying the pretrained classifier. Our approach applies to both the white-box and the black-box settings of the pretrained classifier. We refer to this defense as *denoised smoothing*, and we demonstrate its effectiveness through extensive experimentation on ImageNet and CIFAR-10. Finally, we use our approach to provably defend the Azure, Google, AWS, and ClarifAI image classification APIs. Our code replicating all the experiments in the paper can be found at: `https://github.com/microsoft/denoised-smoothing`[1].

## 1 Introduction

Image classification using deep learning, despite its recent success, is well-known to be susceptible to *adversarial attacks*: small, imperceptible perturbations of the inputs that drastically change the resulting predictions (Szegedy et al., 2013; Goodfellow et al., 2015; Carlini & Wagner, 2017b). To solve this problem, many works proposed heuristic defenses that build models robust to adversarial perturbations, though many of these defenses were broken using more powerful adversaries (Carlini & Wagner, 2017a; Athalye et al., 2018; Uesato et al., 2018). This has led researchers to both strengthen empirical defenses (Kurakin et al., 2016; Madry et al., 2017) as well as to develop *certified* defenses that come with robustness guarantees, i.e., classifiers whose predictions are constant within a neighborhood of their inputs (Wong & Kolter, 2018; Raghunathan et al., 2018a; Cohen et al., 2019; Salman et al., 2019a). However, the majority of these defenses require that the classifier be trained (from scratch) specifically to optimize the robust performance criterion, making the process of building robust classifiers a computationally expensive one.

In this paper, we consider the problem of generating a provably robust classifier *without* retraining the underlying model at all. This problem has not been investigated before, as previous works on provable robustness mainly focus on *training* classifiers for this objective. There are several use cases that make this problem interesting. For example, a provider of a large-scale image classification API may want to offer a "robust" version of the API, but may not want to maintain and/or continually retrain two models that need to be evaluated and validated separately. Even more realistically, a user of a public vision API might want to use that API to create robust predictions (presuming that the API performs well on clean data), but may not have access to the underlying non-robust model. In both cases (which exemplify the white-box and the black-box settings respectively), it would be highly desirable if one could simply apply an off-the-shelf "filter" that would allow practitioners to automatically generate a *provably* robust model from this standard model.

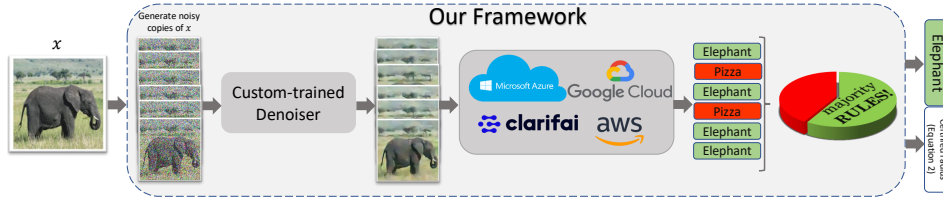

Figure 1: Given a clean image x, our denoised smoothing procedure creates a smoothed classifier by appending a denoiser to any pretrained classifier (e.g. online commercial APIs) so that the pipeline predicts in majority the correct class under Gaussian noise corrupted-copies of x. The resultant classifier is *certifiably* robust against $\ell_2$-perturbations of its input.

Table 1: Certified top-1 accuracy of ResNet-50 on **ImageNet** at various $\ell_2$ radii (Standard accuracy is in parenthesis).

| $\ell_2$ RADIUS (IMAGENET) | 0.25 | 0.5 | 0.75 | 1.0 | 1.25 | 1.5 |
|---|---|---|---|---|---|---|
| COHEN ET AL. (2019) (%) | $^{(70)}62$ | $^{(70)}52$ | $^{(62)}45$ | $^{(62)}39$ | $^{(62)}34$ | $^{(50)}29$ |
| NO DENOISER (BASELINE) (%) | $^{(49)}32$ | $^{(12)}4$ | $^{(12)}2$ | $^{(0)}0$ | $^{(0)}0$ | $^{(0)}0$ |
| OURS (BLACK-BOX) (%) | $^{(69)}48$ | $^{(56)}31$ | $^{(56)}19$ | $^{(34)}12$ | $^{(34)}7$ | $^{(30)}4$ |
| OURS (WHITE-BOX) (%) | $^{(67)}50$ | $^{(60)}33$ | $^{(60)}20$ | $^{(38)}14$ | $^{(38)}11$ | $^{(38)}6$ |

Table 2: Certified accuracy of ResNet-110 on **CIFAR-10** at various $\ell_2$ radii (Standard accuracy is in parenthesis).

| $\ell_2$ RADIUS (CIFAR-10) | 0.25 | 0.5 | 0.75 | 1.0 | 1.25 | 1.5 |
|---|---|---|---|---|---|---|
| COHEN ET AL. (2019) (%) | $^{(77)}59$ | $^{(77)}45$ | $^{(65)}31$ | $^{(65)}21$ | $^{(45)}18$ | $^{(45)}13$ |
| NO DENOISER (BASELINE) (%) | $^{(10)}7$ | $^{(9)}3$ | $^{(9)}0$ | $^{(16)}0$ | $^{(16)}0$ | $^{(16)}0$ |
| OURS (BLACK-BOX) (%) | $^{(81)}45$ | $^{(68)}20$ | $^{(21)}15$ | $^{(21)}13$ | $^{(16)}11$ | $^{(16)}10$ |
| OURS (WHITE-BOX) (%) | $^{(72)}56$ | $^{(62)}41$ | $^{(62)}28$ | $^{(44)}19$ | $^{(42)}16$ | $^{(44)}13$ |

Motivated by this, we propose a new approach to obtain a *provably* robust classifier from a fixed pretrained one, without any additional training or fine-tuning of the latter. This approach is depicted in Figure 1. The basic idea, which we call *denoised smoothing*, is to prepend a custom-trained denoiser before the pretrained classifier, and then apply randomized smoothing (Lecuyer et al., 2018; Li et al., 2018; Cohen et al., 2019). Randomized smoothing is a certified defense that converts any given classifier $f$ into a new smoothed classifier $g$ that is characterized by a non-linear Lipschitz property (Salman et al., 2019a). When queried at a point $x$, the smoothed classifier $g$ outputs the class that is most likely to be returned by $f$ under isotropic Gaussian perturbations of its inputs. Unfortunately, randomized smoothing requires that the underlying classifier $f$ is robust to relatively large random Gaussian perturbations of the input, which is not the case for off-the-shelf pretrained models. By applying our custom-trained denoiser to the classifier $f$, we can effectively *make $f$* robust to such Gaussian perturbations, thereby making it "suitable" for randomized smoothing.

Key to our approach is how we train our denoisers, which is not merely to reconstruct the original image, but also to maintain its original label predicted by $f$. Similar heuristics have been used before; indeed, some of the original adversarial defenses involved applying input transformations to "remove" adversarial perturbations (Guo et al., 2017; Liao et al., 2018; Prakash et al., 2018; Xu et al., 2018), but these defenses were soon broken by more sophisticated attacks (Athalye et al., 2018; Athalye & Carlini, 2018; Carlini & Wagner, 2017a). In contrast, the approach we present here exploits the certified nature of randomized smoothing to ensure that our defense is provably secure.

**Our contribution** is demonstrating, for the first time, a simple yet effective method for converting any pretrained classifier into a provably robust one. This applies both to the setting where we have white-box access to the classifier, *and* to the setting where we only have black-box access. We verify the efficacy of our method through extensive experimentation on ImageNet and CIFAR-10. We are able to convert pretrained ResNet-18/34/50 and ResNet-110, on CIFAR-10 and ImageNet respectively, into certifiably robust models; our results are summarized in Tables 1 and 2 (details are in section 3)[2]. For instance, we are able to boost the certified accuracy of an ImageNet-pretrained ResNet-50 from 4%

to: 31% for the black-box access setting, and 33% for the white-box access setting, under adversarial perturbations with $\ell_2$ norm less than 127/255. We also show the effectiveness of our method through real-world experiments on the Azure, Google, AWS, and ClarifAI image classification APIs. We are able to wrap these vision APIs with our method, leading to provably robust versions of these APIs despite being black-box.

## 2 Denoised Smoothing

In this section, we discuss why randomized smoothing is not, in general, directly effective on off-the-shelf classifiers. Later, we describe our proposed *denoised smoothing* method for solving this problem. We start by introducing some background on randomized smoothing. We refer the reader to Cohen et al. (2019) and Salman et al. (2019a) for a more detailed description of this technique.

### 2.1 Background on Randomized Smoothing

Given a classifier $f$ mapping inputs in $\mathbb{R}^d$ to classes in $\mathcal{Y}$, the randomized smoothing procedure converts the *base* classifier $f$ into a new, *smoothed* classifier $g$. Specifically, for input $x$, $g$ returns the class that is most likely to be returned by the base classifier $f$ under isotropic Gaussian noise perturbations of $x$, i.e.,

$$g(x) = \arg\max_{c \in \mathcal{Y}} \ \mathbb{P}[f(x+\delta) = c] \quad \text{where } \delta \sim \mathcal{N}(0, \sigma^2 I) \ . \tag{1}$$

where the noise level $\sigma$ controls the tradeoff between robustness and accuracy: as $\sigma$ increases, the robustness of the smoothed classifier increases while its standard accuracy decreases.

Cohen et al. (2019) presented a tight robustness guarantee for the *smoothed* classifier $g$ and gave an efficient algorithm based on Monte Carlo sampling for the prediction and certification of $g$. The robustness guarantee of the *smoothed* classifier is based on the Neyman-Pearson lemma (Cohen et al., 2019)[3]. The procedure is as follows: suppose that when the base classifier $f$ classifies $\mathcal{N}(x, \sigma^2 I)$, the class $c_A$ is returned with probability $p_A = \mathbb{P}(f(x+\delta) = c_A)$, and the "runner-up" class $c_B$ is returned with probability $p_B = \max_{c \neq c_A} \mathbb{P}(f(x+\delta) = c)$. The smoothed classifier $g$ is robust around $x$ within the radius

$$R = \frac{\sigma}{2} \left( \Phi^{-1}(p_A) - \Phi^{-1}(p_B) \right), \tag{2}$$

where $\Phi^{-1}$ is the inverse of the standard Gaussian CDF. When $f$ is a deep neural network, computing $p_A$ and $p_B$ accurately is not practical. To mitigate this problem, Cohen et al. (2019) used Monte Carlo sampling to estimate some $\underline{p_A}$ and $\overline{p_B}$ such that $\underline{p_A} \leq p_A$ and $\overline{p_B} \geq p_B$ with arbitrarily high probability. The certified radius is then computed by replacing $p_A, p_B$ with $\underline{p_A}, \overline{p_B}$ in Equation 2.

### 2.2 Image Denoising: a Key Preprocessing Step for Denoised Smoothing

As the above guarantee suggests, randomized smoothing gives a framework for certifying a classifier $f$ without any restrictions on the classifier itself. However, naively applying randomized smoothing on a standard-trained classifier gives very loose certification bounds (as verified by our experiments in section 3). This is because standard classifiers, in general, are not trained to be robust to Gaussian perturbations of their inputs (leading to small $p_A$, hence small $R$, in Equation 2). To solve this problem, previous works use Gaussian noise augmentation (Cohen et al., 2019) and adversarial training (Salman et al., 2019a) to train the base classifier.

We propose *denoised smoothing*, a general method that renders randomized smoothing effective for pretrained models. The goal is to certify existing pretrained classifiers using randomized smoothing without modifying those classifiers, while getting non-trivial certificates. We identify two common scenarios for our method: 1) we have complete knowledge and white-box access to the pretrained classifiers (e.g. API service providers). In this setting, we can back-propagate gradients efficiently through the pretrained classifiers; 2) we only have black-box access to the pretrained classifiers (e.g. API users).

Our method avoids using Gaussian noise augmentation to train the base classifier $f$, but instead, uses an image denoising pre-processing step before passing inputs through $f$. In our setting, denoising is aimed at removing the Gaussian noise used in randomized smoothing. More concretely, we do this by augmenting the classifier $f$ with a custom-trained denoiser $\mathcal{D}_\theta : \mathbb{R}^d \to \mathbb{R}^d$. Thus, our new base classifier is defined as $f \circ \mathcal{D}_\theta : \mathbb{R}^d \to \mathcal{Y}$.

Assuming the denoiser $D_\theta$ is effective at removing Gaussian noise, our framework is characterized to classify well under Gaussian perturbation of its inputs. Our procedure, illustrated in Figure 1, is then formally defined as taking the majority vote using this new base classifier $f \circ \mathcal{D}_\theta$:

$$g(x) = \arg\max_{c \in \mathcal{Y}} \ \mathbb{P}[f(\mathcal{D}_\theta(x + \delta)) = c] \quad \text{where } \delta \sim \mathcal{N}(0, \sigma^2 I) . \tag{3}$$

The *smoothed* classifier $g$ is guaranteed to have fixed prediction within an $\ell_2$ ball of radius $R$ (calculated using Equation 2) centered at $x$. Note that by applying randomized smoothing within our framework, we are robustifying the new classifier $f \circ \mathcal{D}_\theta$, not the old pretrained classifier $f$.

Our defense can be seen as a form of image processing, where we perform input transformations before classification. As opposed to previous works that also used image denoising as an empirical defense (Gu & Rigazio, 2014; Liao et al., 2018; Xie et al., 2019; Gupta & Rahtu, 2019), our method gives provable robustness guarantees. Our denoisers are not intended to remove the adversarial noise, which could lie in some obscure high-dimensional sub-space (Tramèr et al., 2017) and is computationally hard to find (Carlini & Wagner, 2017b). In contrast, our denoisers are only needed to "remove" the Gaussian noise used in randomized smoothing. *In short, we effectively transform the problem of adversarial defense to the problem of Gaussian denoising*; the better the denoising performance, in terms of the custom objectives we will mention shortly, the more robust the resulting smoothed classifier.

We note that Lecuyer et al. (2018) experimented with stacking denoising autoencoders before DNNs as well to scale PixelDP to practical DNNs. This looks similar to our proposed approach. However, our work differs from theirs: 1) in the way these denoisers are trained, and 2) in the fact that we do not finetune the classifiers afterwards whereas Lecuyer et al. (2018) do. Furthermore, their denoising autoencoder was largely intended as a heuristic to speed up training, and differs quite substantially from our application to certify pretrained classifiers.

### 2.3 Training the Denoiser $\mathcal{D}_\theta$

The effectiveness of denoised smoothing highly depends on the denoisers we use. For each noise level $\sigma$, we train a separate denoiser. In this work, we explore two different objectives for training the denoiser $\mathcal{D}_\theta$: 1) the mean squared error objective (MSE), and 2) the stability objective (STAB).

**MSE objective**: this is the most commonly used objective in image denoising. Given an (unlabeled) dataset $\mathcal{S} = \{x_i\}$ of clean images, a denoiser is trained by minimizing the reconstruction objective, i.e., the MSE between the original image $x_i$ and the output of the denoiser $\mathcal{D}_\theta(x_i + \delta)$, where $\delta \sim \mathcal{N}(0, \sigma^2 I)$. Formally, the loss is defined as follows,

$$L_{\text{MSE}} = \mathbb{E}_{\mathcal{S},\delta} \|\mathcal{D}_\theta(x_i + \delta) - x_i\|_2^2 \tag{4}$$

This objective allows for training $\mathcal{D}_\theta$ in an unsupervised fashion. In this work, we focus on the additive white Gaussian noise denoising, which is one of the most studied discriminative denoising models (Zhang et al., 2017, 2018b).

**Stability objective**: the MSE objective turns out not to be the best way to train denoisers for our goal. We want the objective to also take into account the performance of the downstream classification task. However, the MSE objective does not actually optimize for this goal. Thus, we explore another objective that explicitly considers classification under Gaussian noise. Specifically, given a dataset $\mathcal{S} = \{(x_i, y_i)\}$, we train a denoiser $\mathcal{D}_\theta$ from scratch with the goal of classifying images corrupted with Gaussian noise:

$$L_{\text{Stab}} = \mathbb{E}_{\mathcal{S},\delta} \ell_{\text{CE}}(F(\mathcal{D}_\theta(x_i + \delta)), f(x_i)) \quad \text{where } \delta \sim \mathcal{N}(0, \sigma^2 I) , \tag{5}$$

where $F(x)$, that outputs probabilities over the classes, is the *soft* version of the *hard* classifier $f(x)$ (i.e., $f(x) = \arg\max_{c \in \mathcal{Y}} F(x)$), and $\ell_{CE}$ is the cross entropy loss. We call this new objective the *stability objective*[4], and we refer to it as STAB in what follows[5]. This objective can be applied both in the white-box and black-box settings of the pretrained classifier:

- *White-box pretrained classifiers*: since we have white-box access to the pretrained classifier in hand, we can backpropagate gradients through the classifier to optimize the denoisers

using STAB. In other words, we train denoisers from scratch to minimize the classification error using the pseudo-labels given by the pretrained classifier.

- *Black-box pretrained classifiers*: since we only have black-box access to these classifiers, it is difficult to use their gradient information. We get around this problem by using (pretrained) *surrogate classifiers*[6] as proxies for the actual classifiers we are defending. More specifically, we train the denoisers to minimize the stability loss of the surrogate classifiers. It turns out that training denoisers in such a fashion can transfer to unseen classifiers.

We would like to stress that when using the stability objective, we never update the parameters of the underlying classifier neither in the white-box nor the black-box settings. The classifier instead can be seen as providing high-level guidance for training the denoiser.

**Combining the MSE and Stability Objectives**: We explore a hybrid training scheme which connects the low-level image denoising task to high-level image classification. Inspired by the "pretraining+fine-tuning" paradigm in machine learning, denoisers trained with MSE can be good initializations for training with STAB. Therefore, we combine these two objectives by fine-tuning the MSE-denoisers using STAB. We refer to this as STAB+MSE.

In this work, we experiment with all the above mentioned ways of training $\mathcal{D}_\theta$. A complete overview of the objectives used for different classifiers is given in Table 3. Note that we experiment with MSE and STAB+MSE for all the classifiers, but we do STAB-training from scratch only on CIFAR-10 due to computational constraints.

## 3 Experiments

In this section, we present our experiments to robustify pretrained ImageNet and CIFAR-10 classifiers. We use two recent denoisers: DnCNN (Zhang et al., 2017) and MemNet (Tai et al., 2017)[7].

For all experiments, we compare with: 1) certifying pretrained classifiers without stacking any denoisers before them (denoted as *"No denoiser"*), the baseline which we show is inferior to our proposal, and 2) certifying classifiers trained via Gaussian noise augmentation (Cohen et al., 2019), an upper limit of our proposal, as they have the luxury of training the whole classifier but we do not, which is the whole point of our paper.

Table 3: The objectives used in our experiments. Here, "STAB + MSE" means fine-tuning MSE-denoisers with the stability objective. ✓: Conducted experiment, ✗: Potential experiment that is not conducted due to computational constraints.

| CLASSIFIER'S ACCESS TYPE | | DENOISER'S OBJECTIVE | | |
|---|---|---|---|---|
| | | MSE | STAB + MSE | STAB |
| CIFAR-10 | WHITE-BOX | ✓ | ✓ | ✓ |
| | BLACK-BOX | ✓ | ✓ | ✓ |
| IMAGENET | WHITE-BOX | ✓ | ✓ | ✗ |
| | BLACK-BOX | ✓ | ✓ | ✗ |
| VISION-APIS | BLACK-BOX | ✓ | ✓ | ✗ |

→

BETTER DENOISED SMOOTHING

For a given (classifier $f$, denoiser $\mathcal{D}_\theta$) pair, the certified radii of the data points in a given dataset are calculated using Equation 2 with $f \circ \mathcal{D}_\theta$ as the base classifier. The certification curves are then plotted by calculating the percentage of the data points whose radii are larger than a given $\ell_2$-radius. In the following experiments, we only report the results for $\sigma = 0.25$,[8] and we report the best curves over the denoiser architectures mentioned above. For the complete results using other values of $\sigma$, we refer the reader to Appendix B. The compute resources and training time for our experiments are shown in Table 4 in the Appendix. Note that we can train denoisers on different datasets in reasonable time. For more details on the architectures of the classifiers/denoisers, training/certification hyperparameters, etc., we refer the reader to Appendix A.

### 3.1 Certifying White-box Pretrained Classifiers

In this experiment, we assume that the classifier to be defended is known and accessible by the defender, but the defender is not allowed to train or fine-tune this classifier.

**For CIFAR-10**, this classifier is a pretrained ResNet-110 model. The results are shown in Figure 2a. Attaching a denoiser trained on the stability objective (STAB) leads to better certified accuracies

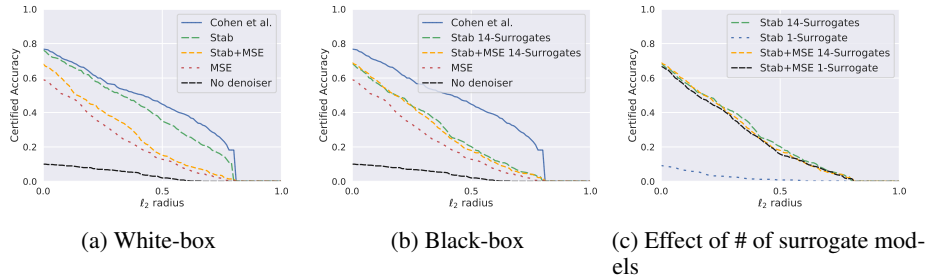

(a) White-box       (b) Black-box       (c) Effect of # of surrogate models

Figure 2: Certifying (a) *white-box* and (b)(c) *black-box* ResNet-110 **CIFAR-10** classifiers using various denoisers. $\sigma = 0.25$.

than attaching a denoiser trained on the MSE objective or only finetuned on the stability objective (STAB+MSE). All of these substantially surpass the "No denoiser" baseline; we achieve an *improvement* of 49% in certified accuracy (over the "No denoiser" baseline) against adversarial perturbations with $\ell_2$ norm less than 64/255 (see Table 2 for more results).

Additionally, Figure 2a plots the certified accuracies of a ResNet-110 classifier trained using Gaussian data augmentation (Cohen et al., 2019). Even though our method does not modify the underlying standard-trained ResNet-110 model (as opposed to Cohen et al. (2019)), our STAB-denoiser achieves similar certified accuracies as the Guassian data-augmentated randomized smoothing model.

**For ImageNet**, we apply our method to PyTorch-pretrained ResNet-18/34/50 classifiers. We assume that we have white-box access to these pretrained models. The results are shown in Figure 3. STAB+MSE performs better than MSE, and again, both of these substantially improve over the "No denoiser" baseline; we achieve an *improvement* of 29% in certified accuracy (over the "No denoiser" baseline) against adversarial perturbations with $\ell_2$ norm less than 127/255 (see Table 1 for more results). Note that we do not experiment with STAB denoisers on ImageNet as it is computationally expensive to train those denoisers, instead we save time by fine-tuning a MSE-denoiser on the stability objective (STAB+MSE)[9].

## 3.2  Certifying Black-box Pretrained Classifiers
In this experiment, we assume that the classifier to be defended is black-box.

**For CIFAR-10**, this classifier is a pretrained ResNet-110 model. The results are shown in Figure 2b. Similar to the white-box access setting, STAB leads to better certified accuracies than both STAB-MSE and MSE. Note that in this setting, STAB and STAB-MSE are both trained with 14 surrogate models (in order to transfer to the black-box ResNet-110). See Appendix A for details of these surrogate models.

It turns out that for STAB-MSE, only one surrogate CIFAR-10 classifier (namely Wide-ResNet-28-10) is also sufficient for the denoiser to transfer well to ResNet-110 as shown in Figure 2c, whereas for STAB, more surrogate models are needed. A detailed analysis of the effect of the number of surrogate models on the performance of STAB and STAB+MSE is deferred to Appendix C.

Overall, we achieve an *improvement* of 38% in certified accuracy (over the "No denoiser" baseline) against adversarial perturbations with $\ell_2$ norm less than 64/255. It turns out the certified accuracies obtained for black-box ResNet-110 are lower than those obtained for Gaussian data-augmented ResNet-110 (see Figure 2 and Table 2), which is expected as we have less information in the black-box access setting.

**For ImageNet**, we again consider PyTorch-pretrained ResNet-18/34/50 classifiers, but now we treat them as "black-box" models. The results are shown in Figure 4, and are similar to the CIFAR-10 results, i.e., attaching a STAB+MSE-denoiser trained on surrogate models leads to a more robust model than attaching a MSE-denoiser. Note that here for STAB+MSE, we fine-tune a MSE-denoiser on the stability objective using *only one* surrogate model due to computational constraints, and also because, as observed on CIFAR-10, the performance of STAB+MSE is similar whether only 1 or 14 surrogate models are used. The exact surrogate models used are shown in Figure 4. For example, in Figure 4c, the black-box model to be defended is ResNet-50, so ResNet-18/34 are used as surrogate models. Note that using either model to fine-tune the denoiser leads to roughly the same certified accuracies. We achieve an *improvement* of 27% in certified accuracy (over the "No denoiser"

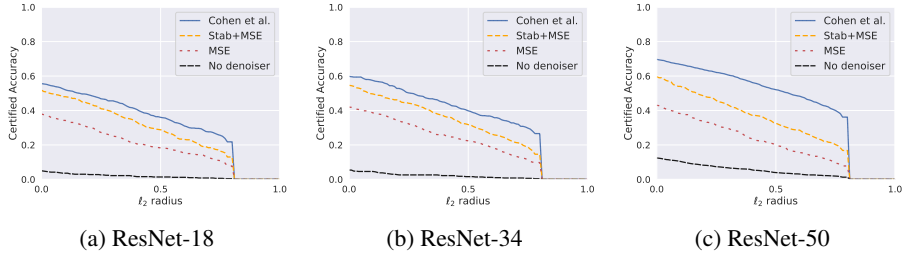

(a) ResNet-18                (b) ResNet-34                (c) ResNet-50

Figure 3: Certifying *white-box* ResNet-18/34/50 **ImageNet** classifiers using various denoisers. $\sigma = 0.25$.

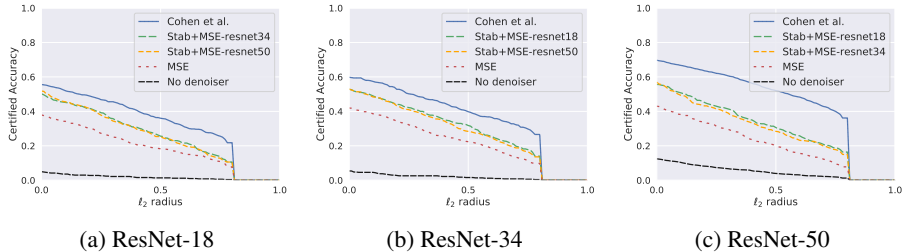

(a) ResNet-18                (b) ResNet-34                (c) ResNet-50

Figure 4: Certifying *black-box* ResNet-18/34/50 **ImageNet** classifiers using various denoisers. $\sigma = 0.25$. Note how fine-tuning the denoisers by attaching surrogate classifiers maintains the high certification accuracy as the white-box setting.

baseline) against adversarial perturbations with $\ell_2$ norm less than 127/255 (see Table 1 for more results).

### 3.3   Perceptual Performance of Denoisers

We note that although the certification results of stability-trained denoisers (STAB and STAB+MSE) are better than the MSE-trained ones, the actual denoising performance of the former does not seem to be as good as the latter. Figure 5 shows an example of denoising a noisy image of an elephant (noise level $\sigma = 1.0$). The reconstructed image using stability-trained denoisers has some strange artifacts. For more examples, see Appendix F.

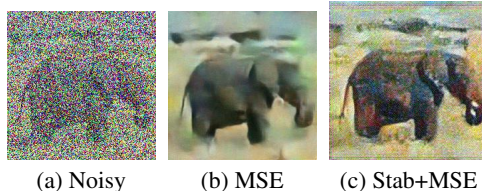

(a) Noisy        (b) MSE        (c) Stab+MSE

Figure 5: Different denoising performance for different denoisers (noise level $\sigma = 1.00$). Note that, although the Stab+MSE denoiser (trained on ResNet-18) leads to strange artifacts as compared to the MSE-denoiser, it gives better certification results as shown in Figure 3.

## 4   Defending Public Vision APIs

We demonstrate that our approach can provide certification guarantees for commercial classifiers. We consider four public vision APIs: Azure Computer Vision[10], Google Cloud Vision[11], AWS Rekognition[12], and Clarifai[13] APIs the models of which are not revealed to the public. Previous works have demonstrated that the Google Cloud Vision API is vulnerable to adversarial attacks (Ilyas et al., 2018; Guo et al., 2019). In this work, we demonstrate for the first time, to the best of our knowledge, a *certifiable* defense for online APIs. Our approach is general and applicable to any API with no knowledge whatsoever of the API's underlying model. Our defense treats the API as a black-box and only requires query-access to it.

We focus on the classification service of each API (for Azure Vision API, this is the image tagging service). Given an input image, each API returns a sorted list of related labels ranked by the corresponding confidence scores. The simplest way to build a classifier from the information is to define the classifier's output as the label with the highest confidence score among the list of labels returned.[14] In this work, we adopt this simple strategy of obtaining a classifier from these APIs.

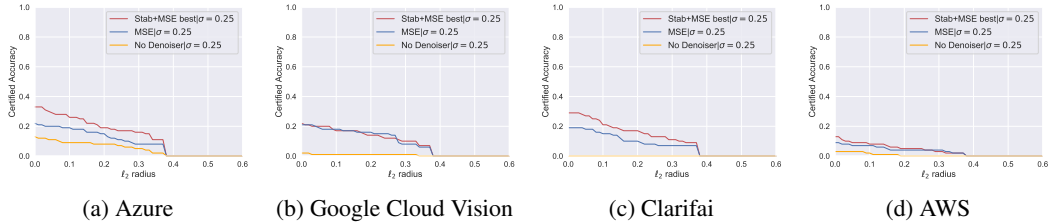

| (a) Azure | (b) Google Cloud Vision | (c) Clarifai | (d) AWS |

Figure 6: Results for certifying four APIs using 100 noise samples per image, $\sigma = 0.25$. "STAB+MSE best" corresponds to the best denoiser out of three denoisers trained via STAB+MSE with three different ImageNet surrogate models: ResNet-18/34/50.

To assess the performance of our method on these APIs, we aggregate 100 random images from the ImageNet validation set and certify their predictions across all four APIs. We use 100 Monte-Carlo samples per data point to estimate the certified radius using Equation 2. We experiment with $\sigma = 0.25$ and with two types of denoisers: an MSE-DnCNN trained on the full ImageNet, and a STAB+MSE DnCNN trained with ResNet-18/34/50 as surrogate models. We also compare to the "No denoiser" baseline, which refers to applying randomized smoothing directly on the APIs.

Figure 6 shows the certified accuracies for all the APIs using both STAB+MSE and MSE. Both denoisers outperform the baseline. Note how the certification results are in general best for stability-trained denoisers, although these were trained on surrogate models, and had no access to underlying models of the vision APIs. For details of the surrogate models used and for more results on these APIs, see Appendix D.

We believe this is only a first step towards certifying public machine learning APIs. Here we restrict ourselves to 100 noisy samples due to budget considerations, however, one can always obtain a larger certified radius by querying more noisy samples (e.g. 1k or 10k) (as verified in Appendix D). Our results also suggest the possibility for machine learning service providers to offer robust versions of their APIs without having to change their pretrained classifiers. This can be done by simply wrapping their APIs with *our* custom-trained denoisers.

# 5   Defending Against General $\ell_p$ Threat Models

Throughout the paper, we focus on defending against $\ell_2$ adversarial perturbations, although nothing in theory prevents our method from being applied to other $\ell_p$ threat models. In fact, as long as randomized smoothing works for other $\ell_p$ norms (which has been shown in recent papers (Li et al., 2019; Lee et al., 2019; Dvijotham et al., 2020)), our method automatically works. The only change would be the way our denoisers are trained; instead of training via *Gaussian noise augmentation*, the denoisers shall be trained on data corrupted with noise that is sampled from other distributions (e.g. uniform distribution for $\ell_1$ threat models (Yang et al., 2020)). We emphasize that since our approach utilizes randomized smoothing, any practical or theoretical limitation of randomized smoothing passes on to our approach. For example, Yang et al. (2020) showed that randomized smoothing cannot achieve nontrivial certified accuracy against perturbations of $\ell_p$-norm $\Omega(\min(1, d^{\frac{1}{p} - \frac{1}{2}}))$ when the input dimension d is large. The theoretical result holds in our framework by construction.

# 6   Related Work

In the past few years, numerous defenses have been proposed to build adversarially robust classifiers. In this paper, we distinguish between *robust training based defenses* and *input transformation based defenses*. Although we use randomized smoothing, our defense largely fits into the latter category, as it is based upon a denoiser applied before the pretrained classifier.

## 6.1   Robust Training Based Defenses

Many defenses train a robust classifier via a robust training procedure, i.e., the classifier is trained, usually from scratch, specifically to optimize a robust performance criterion. We characterize two directions of robust training based defenses: empirical defenses and certified defenses.

**Empirical defenses** are those which have been empirically shown to be robust to existing adversarial attacks. The best empirical defense to date is *adversarial training* (Kurakin et al., 2016; Madry et al., 2017), in which a robust classifier is learned by training directly on adversarial examples generated by various threat models (Carlini & Wagner, 2017a; Laidlaw & Feizi, 2019; Wong et al., 2019; Hu et al.,

2020). Although such defenses have been shown to be strong, nothing guarantees that a stronger, not-yet-known, attack would not "break" them. In fact, most empirical defenses proposed in the literature were later broken by stronger adversaries (Carlini & Wagner, 2017a; Athalye et al., 2018; Uesato et al., 2018; Athalye & Carlini, 2018). To put an end to this arms race, a few works tried to build certified defenses that come with formal robustness guarantees.

**Certified defenses** provide guarantees that for any input $x$, the classifier's prediction is constant within a neighborhood of $x$. The majority of the training-based certified defenses rely on minimizing an upper bound of a loss function over all adversarial perturbations (Wong & Kolter, 2018; Wang et al., 2018a,b; Raghunathan et al., 2018a,b; Wong et al., 2018; Dvijotham et al., 2018b,a; Croce et al., 2018; Salman et al., 2019b; Gehr et al., 2018; Mirman et al., 2018; Singh et al., 2018; Gowal et al., 2018; Weng et al., 2018; Zhang et al., 2018a). However, these defenses are, in general, not scalable to large models (e.g. ResNet-50) and datasets (e.g. ImageNet). More recently, a more scalable approach called *randomized smoothing* was proposed as a probabilistically certified defense. Randomized smoothing converts any given classifier into another provably robust classifier by convolving the former with an isotropic Gaussian distribution. It was proposed by several works (Liu et al., 2018; Cao & Gong, 2017) as a heuristic defense without proving any guarantees. A few works afterwards were able to provide formal guarantees for randomized smoothing (Lecuyer et al., 2018; Li et al., 2019; Cohen et al., 2019).

Although, in theory, randomized smoothing does not require any training of the original classifier, in order to get non-trivial robustness results, the original classifier has to be custom-trained from scratch as shown in several papers (Lecuyer et al., 2018; Cohen et al., 2019; Salman et al., 2019a; Zhai et al., 2020; Yang et al., 2020). Lecuyer et al. (2018) experimented with stacking denoising autoencoders before deep neural networks (DNNs) to scale PixelDP to practical DNNs that are tedious to train from scratch. However, there are two key differences between this work and ours: 1) Lecuyer et al. (2018) trained the denoising autoencoder with only the reconstruction loss, as opposed to the classification-based stability loss that we discuss shortly; and 2) this past work further fine-tuned the classifier itself, whereas the central motivation of our paper is to avoid this step. Indeed, the denoising autoencoder in this prior work was largely intended as a heuristic to speed up training, and differs quite substantially from our application to certify pretrained classifiers.

## 6.2 Input Transformation Based Defenses

These defenses try to remove the adversarial perturbations from the input by transforming the input before feeding it to the classifier. Many such defenses have been proposed (but later broken) in previous works (Guo et al., 2017; Meng & Chen, 2017; Xu et al., 2018; Liao et al., 2018). Guo et al. (2017) proposed to use traditional image processing, e.g. image cropping, rescaling, and quilting. Meng & Chen (2017) trained an autoencoder to reconstruct clean images. Xu et al. (2018) used color bit depth reduction and spatial smoothing to reduce the space of adversarial attacks. Liao et al. (2018) trained a classification-guided denoiser to remove adversarial noise. However, all these defenses were broken by stronger attacks (Warren et al., 2017; Carlini & Wagner, 2017a; Athalye et al., 2018; Athalye & Carlini, 2018). To the best of our knowledge, all existing input transformation based defenses are empirical defenses. In this work, we present the first input transformation based defense that provides *provable* guarantees.

## 7 Conclusion and Future Work

In this paper, we presented a simple defense called *denoised smoothing* that can convert existing pretrained classifiers into provably robust ones without any retraining or fine-tuning. We achieve this by prepending a custom-trained denoiser to these pretrained classifers. We experimented with different strategies for training the denoiser and obtained significant boosts over the trivial application of randomized smoothing on pretrained classifiers. We are the first, to the best of our knowledge, to show that we can provably defend online vision APIs.

This is only a first *stab* at provably robustifying pretrained classifiers, and we believe there is still plenty of room for improving our method. For example, training denoisers that can get as good certified accuracies as Cohen et al. (2019) is something that we could not achieve in our paper. We are able to achieve almost as good results as Cohen et al. (2019) in the white-box setting, but not in the black-box setting as shown in Figures 2b and 4. Finding methods that can train denoisers to close the gap between our method and Cohen et al. (2019) remains a valuable future direction.

## Statement of Broader Impact

The vulnerability of deep learning to adversarial attacks poses serious problems in many safety critical computer vision applications, including self-driving cars. Our method is a step towards building robust models that can be deployed reliably in the real world. Using our method, any off-the-shelf classifier, for the first time, can immediately be wrapped by our framework and converted into a provably robust one as we demonstrate on four real world online APIs. Furthermore, one major use case of our method is when users of public APIs want to get provably robust predictions from these APIs, but can not really change or retrain these APIs as they are black-box for them. Our method helps these users simply convert these APIs into provably robust ones by wrapping them with our framework. We believe this is an important step towards having machine learning confidently and safely used and deployed.

## Acknowledgments and Disclosure of Funding

Work supported in part by the Microsoft Corporation and the Bosch Center for AI.

## Footnotes

[1]Please see `http://arxiv.org/abs/2003.01908` for the full and most recent version of this paper.

[2]Tables for ResNet-18/34 on ImageNet are in Appendix B.

[3]This guarantee can also be obtained alternatively by explicitly computing the Lipschitz constant of the smoothed classifier as shown in Salman et al. (2019a); Yang et al. (2020).

[4]Note that the stability objective used here is similar in spirit to, but different in context from, the stability training used in Zheng et al. (2016) and Li et al. (2019)

[5]We also experiment with *classification objective* which uses true label $y_i$ instead of $f(x_i)$. Details are shown in Appendix E.

[6]For ImageNet, we experiment with standard-trained ResNet-18/34/50 as surrogate models. For CIFAR-10, we experiment with 14 standard-trained models listed in Appendix A.

[7]We also experiment with DnCNN-wide, a wide version of DnCNN which we define (more details in Appendix A). Note that we do not claim these are the best denoisers in the literature. We just picked two common denoisers. A better choice of denoiser might lead to improvements in our results.

[8]We use the *same* $\sigma$ for training the denoiser and certifying the denoised classifier via Equation 2.

[9]See Table 4 for details on the computation time and resources.

[10]https://docs.microsoft.com/en-us/azure/cognitive-services/computer-vision/

[11]https://cloud.google.com/vision/

[12]https://aws.amazon.com/rekognition/

[13]https://www.clarifai.com/

[14]For AWS, there can be several top labels with same scores. We let the first in the list to be the output.

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
