[Supplementary Material]

# A  Experiments Details

In this appendix, we include details of all the experiments conducted in our paper.

## A.1  Denoiser Models

The denoisers used in this paper are:

1. DnCNN (Zhang et al., 2017) https://github.com/cszn/DnCNN/tree/master/TrainingCodes/dncnn_pytorch.

2. DnCNN-Wide: a wide version of the DnCNN architecture; specifically convolutional layers with a width of 128 instead of 64. Check the code for more details.

3. MemNet (Tai et al., 2017) https://github.com/tyshiwo/MemNet.

We run experiments with all of these denoisers on CIFAR-10. For ImageNet , we stick to DnCNN only due to GPU memory constraints. Note that we do not claim these are the best denoisers in the literature. We just picked these common denoisers. A better choice of denoisers might lead to improvements in our results.

**Training details**  Here we include the training details of each denoiser used in our paper. The results in the paper are reported for the best denoisers over all the hyperparameters (architectures, optimizer, and learning rate) summarized in the following table. Also, for each architecture, the reported training time of each model is averaged over all the instances of training this architecture with various optimizers and learning rates.

Table 4: Average training statistics for our denoisers. We run our experiments on NVIDIA P100 and V100 GPUs. For STAB+MSE, the "+" sign refers to the additional epochs/total time incurred over the MSE baseline since STAB+MSE is basically fine-tuning MSE denoisers.

| TRAINED DENOISER | OBJECTIVE | COMPUTE | #EPOCHS | OPTIMIZER | LEARNING RATE | SEC/EPOCH | TOTAL TIME (HR) |
|---|---|---|---|---|---|---|---|
| CIFAR-10/DNCNN | MSE | $1 \times$ P100 | 90 | ADAM | $1e-3$ | 31 | 0.78 |
| | STAB+MSE RESNET-110 | $1 \times$ P100 | +20 | {ADAM, SGD} | $\{1e-4, 1e-5\}$ | 57 | +0.32 |
| | STAB RESNET-110 | $1 \times$ P100 | 600 | ADAMTHENSGD | SEE BELOW | 59 | 9.80 |
| CIFAR-10/DNCNN-WIDE | MSE | $1 \times$ P100 | 90 | ADAM | $1e-3$ | 122 | 3.05 |
| | STAB+MSE RESNET-110 | $1 \times$ P100 | +20 | {ADAM, SGD} | $\{1e-4, 1e-5\}$ | 135 | +0.75 |
| | STAB RESNET-110 | $1 \times$ P100 | 600 | ADAMTHENSGD | SEE BELOW | 156 | 26.00 |
| CIFAR-10/MEMNET | MSE | $1 \times$ P100 | 90 | ADAM | $1e-3$ | 85 | 2.13 |
| | STAB+MSE RESNET-110 | $1 \times$ P100 | +20 | {ADAM, SGD} | $\{1e-4, 1e-5\}$ | 118 | +0.66 |
| | STAB RESNET-110 | $1 \times$ P100 | 600 | ADAMTHENSGD | SEE BELOW | 125 | 20.83 |
| IMAGENET/DNCNN | MSE | $4 \times$ V100 | 5 | ADAM | $1e-4$ | 6320 | 8.78 |
| | STAB+MSE RESNET-18 | $4 \times$ V100 | +20 | ADAM | $1e-5$ | 6500 | +36.11 |
| | STAB+MSE RESNET-34 | $4 \times$ V100 | +20 | ADAM | $1e-5$ | 6900 | +38.33 |
| | STAB+MSE RESNET-50 | $4 \times$ V100 | +20 | ADAM | $1e-5$ | 7620 | +42.33 |
| VISION-APIS/DNCNN | SAME AS IMAGENET | | | SAME AS IMAGENET | | | |

Note that for STAB training, we find that using ADAMTHENSGD leads to significantly better performance than using only one of them. For this setting, we basically use ADAM with a learning rate of $1e-3$ for 50 epochs, then use SGD with the following settings:

1. SGD with learning rate that starts at $1e-4$ and drops by a factor of 10 every 100 epochs.

2. SGD with learning rate that starts at $1e-3$ and drops by a factor of 10 every 100 epochs.

3. SGD with learning rate that starts at $1e-4$ and drops by a factor of 10 every 200 epochs.

4. SGD with learning rate that starts at $1e-3$ and drops by a factor of 10 every 200 epochs.

5. SGD with learning rate that starts at $1e-2$ and drops by a factor of 10 every 200 epochs.

6. SGD with learning rate that starts at $1e-3$ and drops by a factor of 10 every 400 epochs.

Also, note that for STAB+MSE RESNET-110, we sweep over the *product* of the two sets of optimizers {ADAM, SGD}, and learning rates $\{1e-4, 1e-5\}$.

Please refer to our code for further details!

## A.2 Pretrained Classifiers

Our method presented in the paper works on any pretrained classifier. For the sake of demonstrating its effectiveness, we use standard CIFAR-10 and ImageNet neural network architectures.

**On CIFAR-10**, we train our own versions of the classifiers found in the following repository `https://github.com/kuangliu/pytorch-cifar`, namely, we train:

• ResNet-110 • Wide-ResNet-28-10 • Wide-ResNet-40-10 • VGG-16 • VGG-19 • ResNet-18 • Pre-ActResNet-18 • GoogLeNet • DenseNet-121 • ResNeXt29_2x64d • MobileNet • MobileNet-V2 • SENet-18 • ShuffleNet-V2 • EfficientNet-B0.[15]

We train these classifiers in a standard way with data augmentation (random horizontal flips and random crops). We train each model for 300 epochs using SGD with an initial learning rate of 0.1 that drops by a factor of 10 each 100 epochs. We provide these pretrained models and code to train them in the repository accompanying this paper.

**On ImageNet**, we use PyTorch's ResNet-18, ResNet-34, and ResNet-50 pretrained ImageNet models from the following link `https://pytorch.org/docs/stable/torchvision/models.html`.

## A.3 Certification Details

In order to certify our (denoiser, classifier) pairs, we use the CERTIFY randomized smoothing algorithm of Cohen et al. (2019).

For CERTIFY, unless otherwise specified, we use $n = 10,000$, $n_0 = 100$, $\alpha = 0.001$. Note that in (Cohen et al., 2019), $n = 100,000$, which leads to better certification results. Due to computational constraints, we decrease this by a factor of 10. All our results can be improved by increasing $n$.

In all the above settings, we report the best models over all the hyperparameters we mentioned.

## A.4 Source code

Our code and trained denoisers and classifiers can be found in the the following GitHub repository: `https://github.com/microsoft/denoised-smoothing`. The repository also includes all our training and certification logs, which allows for easy replication of all our results!

# B    Detailed Experimental Results

In this part, we show more detailed experimental results of our method on ImageNet and CIFAR-10 (More pretrained classifiers and more noise levels).

## B.1    Our Best Certified Accuracies over a Range of $\ell_2$-Radii

Here we show the best certified accuracy we get at various $\ell_2$-radii. The results are shown in Table 5, Table 6, Table 7 and Table 8. Note that in both white-box access and black-box access settings, we outperform the baseline without denoisers[16].

Table 5: Certified top-1 accuracy of **ResNet-50** on **ImageNet** at various $\ell_2$ radii (Standard accuracy is in parenthesis).

| $\ell_2$ RADIUS (IMAGENET) | 0.25 | 0.5 | 0.75 | 1.0 | 1.25 | 1.5 |
|---|---|---|---|---|---|---|
| COHEN ET AL. (2019) (%) | $^{(70)}62$ | $^{(70)}52$ | $^{(62)}45$ | $^{(62)}39$ | $^{(62)}34$ | $^{(50)}29$ |
| NO DENOISER (BASELINE) (%) | $^{(49)}32$ | $^{(12)}4$ | $^{(12)}2$ | $^{(0)}0$ | $^{(0)}0$ | $^{(0)}0$ |
| OURS (BLACK-BOX) (%) | $^{(69)}48$ | $^{(56)}31$ | $^{(56)}19$ | $^{(34)}12$ | $^{(34)}7$ | $^{(30)}4$ |
| OURS (WHITE-BOX) (%) | $^{(67)}50$ | $^{(60)}33$ | $^{(60)}20$ | $^{(38)}14$ | $^{(38)}11$ | $^{(38)}6$ |

Table 6: Certified top-1 accuracy of **ResNet-34** on **ImageNet** at various $\ell_2$ radii (Standard accuracy is in parenthesis).

| $\ell_2$ RADIUS (IMAGENET) | 0.25 | 0.5 | 0.75 | 1.0 | 1.25 | 1.5 |
|---|---|---|---|---|---|---|
| COHEN ET AL. (2019) (%) | $^{(60)}50$ | $^{(53)}44$ | $^{(53)}39$ | $^{(53)}33$ | $^{(53)}28$ | $^{(42)}22$ |
| NO DENOISER (BASELINE) (%) | $^{(44)}26$ | $^{(5)}2$ | $^{(5)}1$ | $^{(0)}0$ | $^{(0)}0$ | $^{(0)}0$ |
| OURS (BLACK-BOX) (%) | $^{(65)}47$ | $^{(53)}32$ | $^{(53)}18$ | $^{(34)}12$ | $^{(34)}8$ | $^{(34)}3$ |
| OURS (WHITE-BOX) (%) | $^{(64)}47$ | $^{(55)}32$ | $^{(55)}19$ | $^{(35)}12$ | $^{(35)}8$ | $^{(16)}4$ |

Table 7: Certified top-1 accuracy of **ResNet-18** on **ImageNet** at various $\ell_2$ radii (Standard accuracy is in parenthesis).

| $\ell_2$ RADIUS (IMAGENET) | 0.25 | 0.5 | 0.75 | 1.0 | 1.25 | 1.5 |
|---|---|---|---|---|---|---|
| COHEN ET AL. (2019) (%) | $^{(56)}47$ | $^{(48)}36$ | $^{(48)}31$ | $^{(48)}26$ | $^{(35)}22$ | $^{(35)}19$ |
| NO DENOISER (BASELINE) (%) | $^{(37)}18$ | $^{(5)}1$ | $^{(5)}1$ | $^{(0)}0$ | $^{(0)}0$ | $^{(0)}0$ |
| OURS (BLACK-BOX) (%) | $^{(60)}42$ | $^{(50)}26$ | $^{(50)}14$ | $^{(28)}7$ | $^{(28)}5$ | $^{(28)}3$ |
| OURS (WHITE-BOX) (%) | $^{(61)}42$ | $^{(52)}29$ | $^{(52)}16$ | $^{(35)}10$ | $^{(35)}6$ | $^{(35)}4$ |

Table 8: Certified accuracy of **ResNet-110** on **CIFAR-10** at various $\ell_2$ radii (Standard accuracy is in parenthesis).

| $\ell_2$ RADIUS (CIFAR-10) | 0.25 | 0.5 | 0.75 | 1.0 | 1.25 | 1.5 |
|---|---|---|---|---|---|---|
| COHEN ET AL. (2019) (%) | $^{(77)}59$ | $^{(77)}45$ | $^{(65)}31$ | $^{(65)}21$ | $^{(45)}18$ | $^{(45)}13$ |
| NO DENOISER (BASELINE) (%) | $^{(10)}7$ | $^{(9)}3$ | $^{(9)}0$ | $^{(16)}0$ | $^{(16)}0$ | $^{(16)}0$ |
| OURS (BLACK-BOX) (%) | $^{(81)}45$ | $^{(68)}20$ | $^{(21)}15$ | $^{(21)}13$ | $^{(16)}11$ | $^{(16)}10$ |
| OURS (WHITE-BOX) (%) | $^{(72)}56$ | $^{(62)}41$ | $^{(62)}28$ | $^{(44)}19$ | $^{(42)}16$ | $^{(44)}13$ |

## B.2 Certifying White-box CIFAR-10 Classifiers

Figure 7 shows the complete results for certifying white-box CIFAR-10 pretrained classifiers at various noise levels $\sigma \in \{0.12, 0.25, 0.50, 1.00\}$. We can see that using denoisers trained with stability objective (STAB) can get close certification results to Cohen et al. (2019). The results for $\sigma = 0.25$ are the same as Figure 2a in the main text.

(a) $\sigma = 0.12$        (b) $\sigma = 0.25$

(c) $\sigma = 0.50$        (d) $\sigma = 1.00$

Figure 7: Results for certifying a ***white-box*** ResNet-110 **CIFAR-10** classifier using various methods.

## B.3 Certifying Black-box CIFAR-10 Classifiers

Figure 8 shows the full results for certifying black-box CIFAR-10 pretrained classifiers at various noise levels $\sigma \in \{0.12, 0.25, 0.50, 1.00\}$. Compared the white-box access setting, there is still a noticeable gap between our method and Cohen et al. (2019). The results for $\sigma = 0.25$ are the same as Figure 2b in the main text. Note that for $\sigma = 0.50$, MSE is better than STAB for small $\ell_2$-radii, but for this range of radii, one would practically choose models with smaller noise level, say $\sigma = 0.12$, as the certified accuracies of the latter are higher in this range of radii.

(a) $\sigma = 0.12$        (b) $\sigma = 0.25$

(c) $\sigma = 0.50$        (d) $\sigma = 1.00$

Figure 8: Results for certifying a ***black-box*** ResNet-110 **CIFAR-10** classifier using various methods.

## B.4   Certifying White-box Access ImageNet classifiers

Figure 9, Figure 10 and Figure 11 show the complete results for certifying white-box ImageNet pretrained classifiers at various noise levels $\sigma \in \{0.25, 0.50, 1.00\}$. The results for $\sigma = 0.25$ are the same as Figure 3 in the main text. Notice that STAB+MSE is better than MSE for all cases. It can be seen that with larger noise level $\sigma$, the gap between our method and Cohen et al. (2019) becomes larger.

(a) $\sigma = 0.25$   (b) $\sigma = 0.50$   (c) $\sigma = 1.00$

Figure 9: Results for certifying a *white-box* **ResNet-18 ImageNet** classifier.

(a) $\sigma = 0.25$   (b) $\sigma = 0.50$   (c) $\sigma = 1.00$

Figure 10: Results for certifying a *white-box* **ResNet-34 ImageNet** classifier.

(a) $\sigma = 0.25$   (b) $\sigma = 0.50$   (c) $\sigma = 1.00$

Figure 11: Results for certifying a *white-box* **ResNet-50 ImageNet** classifier.

## B.5 Certifying Black-box ImageNet classifiers

Figure 12, Figure 13 and Figure 14 show the full results for certifying black-box ImageNet pretrained classifiers at various noise levels $\sigma \in \{0.25, 0.50, 1.00\}$. The results for $\sigma = 0.25$ are the same as Figure 4 in the main text. In general, STAB+MSE outperforms MSE. Notice that similar to the case of white-box access setting, the gap becomes larger between our method and Cohen et al. (2019).

(a) $\sigma = 0.25$       (b) $\sigma = 0.50$       (c) $\sigma = 1.00$

Figure 12: Results for certifying a *black-box* **ResNet-18** ImageNet classifier.

(a) $\sigma = 0.25$       (b) $\sigma = 0.50$       (c) $\sigma = 1.00$

Figure 13: Results for certifying a *black-box* **ResNet-34** ImageNet classifier.

(a) $\sigma = 0.25$       (b) $\sigma = 0.50$       (c) $\sigma = 1.00$

Figure 14: Results for certifying a *black-box* **ResNet-50** ImageNet classifier.

# C More Surrogate Models, Better Transfer

Here we demonstrate that when certifying black-box pretrained classifiers, we can get better certification results if we use more surrogate models when training the denosiers. The comparisons are shown in Figure 15 on CIFAR-10 for STAB and STAB+MSE. We can see that in the case of STAB, using 14 surrogate models is important to generalize to an unseen ResNet-110 classifier especially for large $\sigma$. The results for $\sigma = 0.25$ are the same as Figure 2c in the main text.

(a) $\sigma = 0.12$

(b) $\sigma = 0.25$

(c) $\sigma = 0.50$

(d) $\sigma = 1.00$

Figure 15: Results for certifying a ***black-box*** ResNet-110 **CIFAR-10** classifier using various number of surrogate models with STAB and STAB+MSE.

# D Vision APIs Detailed Results

In this appendix, we present more detailed results of denoised smoothing on the four Vision APIs we consider in this paper.

## D.1 Comparison between Stab+MSE and MSE objectives

Figure 16, Figure 17, Figure 18 and Figure 19 show the comparison between the certification results of STAB+MSE and MSE objectives for various vision APIs, surrogate models (ResNet-18/34/50), and noise levels $\sigma \in \{0.12, 0.25\}$. Observe that the performance of the STAB+MSE objective either roughly matches or outperforms the MSE objective[17].

| (a) ResNet-18 | (b) ResNet-34 | (c) ResNet-50 |

Figure 16: The certification results of the **Azure API** using STAB+MSE (across various surrogate models) and MSE denoisers.

| (a) ResNet-18 | (b) ResNet-34 | (c) ResNet-50 |

Figure 17: The certification results of the **Google Cloud Vision API** using STAB+MSE (across various surrogate models) and MSE denoisers.

| (a) ResNet-18 | (b) ResNet-34 | (c) ResNet-50 |

Figure 18: The certification results of the **Clarifai API** using STAB+MSE (across various surrogate models) and MSE denoisers.

| (a) ResNet-18 | (b) ResNet-34 | (c) ResNet-50 |

Figure 19: The certification results of the **AWS API** using STAB+MSE (across various surrogate models) and MSE denoisers.

## D.2 More Monte Carlo Samples, More Robustness

Here we empirically demonstrate that by using more Monte Carlo samples per image in denoised smoothing, we get better certification bounds. Figure 20 reports the certified accuracies (over 100 samples of the ImageNet validation set), after applying our method to the Azure Vision API, over a range of $\ell_2$-radii, and with 1000 vs. 100 Monte Carlo samples. Indeed, we notice that more samples lead to higher certified accuracies (i.e. more robust versions of the Azure API).

(a) Azure API

Figure 20: The certification results, with 1000 vs. 100 Monte Carlo samples per image, of the **Azure API** with an MSE-denoiser.

# E    Stability vs Classification Objectives

The stability objective, introduced in subsection 2.3, is similar to another objective, namely the classification objective, which we will refer to as CLF. For this objective the loss function, previously defined in Equation 5, is now defined as the cross entropy loss between the output probabilities and the **true labels** $y_i$'s (as opposed to the pseudo-labels $f(x_i)$'s generated by the pretrained classifier $f$):

$$L_{\text{Clf}} = \mathop{\mathbb{E}}_{\mathcal{S},\delta} \ell_{\text{CE}}(F(\mathcal{D}_\theta(x_i + \delta)), y_i) \tag{6}$$

$$\text{where } \delta \sim \mathcal{N}(0, \sigma^2 I) ,$$

Note that in the main text, we stick with the stability objective, since, as we show in the following subsections, the performance of the stability objective is comparable to that of the classification objective (and sometimes slightly better).

Also note that, for training the denoisers with classification objectives, we use the same hyperparameters as the stability objective shown in Table 4.

## E.1    CIFAR-10

Figure 21 compares the performance of denoisers trained with STAB against denoisers trained with CLF on CIFAR-10. (a) and (b) show the white-box access and black-box access settings, respectively. Observe that the performance of the classification objectives is comparable to that of the stability objective.

(a) White-box                     (b) Black-box

Figure 21: Comparison between stability objective and classification objective on a ResNet-110 **CIFAR-10** classifier.

## E.2    ImageNet

Figure 22 compares MSE-trained denoisers fine-tuned on the stability objective (STAB+MSE) against MSE-trained denoisers fine-tuned on the classification objective (CLF+MSE) on white-box ImageNet ResNet-18/30/50. Again, the performance of the classification objectives is comparable to that of the stability objective.

(a) ResNet-18                (b) ResNet-34                (c) ResNet-50

Figure 22: Comparison between stability objective and classification objective on ResNet-18/34/50 **ImageNet** classifiers.

## E.3 Vision APIs

Figure 23, Figure 24, Figure 25 and Figure 26 show the performance of fine-tuning using the stability objective (STAB+MSE) and fine-tuning using the classification objective (CLF+MSE) on four vision APIs, with $\sigma \in \{0.12, 0.25\}$. We notice that STAB+MSE and CLF+MSE have largely the same performance.

| (a) ResNet-18 | (b) ResNet-34 | (c) ResNet-50 |

Figure 23: The certification results of the **Azure API** with denoisers trained with STAB+MSE vs. CLF+MSE.

| (a) ResNet-18 | (b) ResNet-34 | (c) ResNet-50 |

Figure 24: The certification results of the **Google API** with denoisers trained with STAB+MSE vs. CLF+MSE.

| (a) ResNet-18 | (b) ResNet-34 | (c) ResNet-50 |

Figure 25: The certification results of the **Clarifai API** with denoisers trained with STAB+MSE vs. CLF+MSE.

| (a) ResNet-18 | (b) ResNet-34 | (c) ResNet-50 |

Figure 26: The certification results of the **AWS API** with denoisers trained with STAB+MSE vs. CLF+MSE.

# F   Denoising Examples on ImageNet

Noisy Images

MSE

STAB+MSE on ResNet-18

STAB+MSE on ResNet-34

STAB+MSE on ResNet-50

Figure 27: Performance of the various ImageNet denoisers on noisy images (first row) of standard deviation of 0.12, 0.25, 0.5, and 1.0 respectively from left to right.

Noisy Images

MSE

STAB+MSE on ResNet-18

STAB+MSE on ResNet-34

STAB+MSE on ResNet-50

Figure 28: Performance of the various ImageNet denoisers on noisy images (first row) of standard deviation of 0.12, 0.25, 0.5, and 1.0 respectively from left to right.

Noisy Images

MSE

STAB+MSE on ResNet-18

STAB+MSE on ResNet-34

STAB+MSE on ResNet-50

Figure 29: Performance of the various ImageNet denoisers on noisy images (first row) of standard deviation of 0.12, 0.25, 0.5, and 1.0 respectively from left to right.

Noisy Images

MSE

STAB+MSE on ResNet-18

STAB+MSE on ResNet-34

STAB+MSE on ResNet-50

Figure 30: Performance of the various ImageNet denoisers on noisy images (first row) of standard deviation of 0.12, 0.25, 0.5, and 1.0 respectively from left to right.

Noisy Images

MSE

STAB+MSE on ResNet-18

STAB+MSE on ResNet-34

STAB+MSE on ResNet-50

Figure 31: Performance of the various ImageNet denoisers on noisy images (first row) of standard deviation of 0.12, 0.25, 0.5, and 1.0 respectively from left to right.

## Footnotes

[15]Note that when we train with STAB or STAB+MSE in the **black-box access setting** in the main paper, we use all these models as surrogate models, excluding ResNet-110, since this is the pretrained model that we assume we only have black-box access to.

[16]Table 5 and Table 8 are the same as Table 1 and Table 2 in the main paper, respectively.

[17]Figure 6 in the main text is generated by plotting, for each API, the best certification curve for STAB+MSE across the three surrogate models presented here, along with the MSE curve, and for $\sigma = 0.25$.