[Reviews · NeurIPS 2020]

Review 1

Summary and Contributions: The paper proposes a denoising based adversarial defense method. The custom trained denoiser is applied before passing an image to the target classifier to remove any malicious noise.

Strengths: The problem addressed in the paper is interesting, and the solution to handle such adversarial attacks is impressive.

Weaknesses: The paper has several weaknesses ranging from the algorithm to experimental evaluation. 1. The algorithm proposed looks merely like plug and play from the different existing algorithms such as Cohen et al., Salman et al., Li et al. Therefore, the authors need to define their contribution clearly. Reading the paper looks like the only difference from several existing works is not the training of the target classifier. 2. The claim on line 28-29 that this is the first work is entirely wrong. Several works have been done in the literature that is based on compression, randomization, and mitigation, where the authors have proposed a separate pipeline for adversarial effect removal. Some of the works have also mentioned on page 2, line 53 by the authors. 3. The understanding of the white-box scenario is misleading. The authors must show that the proposed methodology is secure even when the attacker has access to the denoiser. 4. The comparison with Cohen et al. shows that the proposed algorithm is far behind being useful in this crucial direction, or it might become another existing defense. 5. While the authors have claimed that the proposed algorithm can handle the l_p norm attack, the experimental setup is extremely unclear. What all possible existing attacks the algorithm can handle? 6. Are improvements statistically significant?

Correctness: No proper theoritical justification is provided.

Clarity: The paper is tough to follow and read. The authors need to provide significant details in the paper itself. To look at everything in the supplementary or appendix makes the document difficult to read. Please check the formatting as well.

Relation to Prior Work: The related work section is weak and needs significant improvement.

Reproducibility: No

Additional Feedback: The authors need to explain how the denoiser is trained and the parameters that can affect its performance. Post rebuttal: The explanation towards the sensitivity against unseen noise is not provided. Is the denoiser capable in handling unseen noise variation such as trained on gaussian (l2) and tested on laplace. In the rebuttal authors pointed towards training using each category of noise to handle it. I think this can lead to limitation to practial implementation. The contribution against multiple existing works is not properly discussed/justified including experimental gains. The discussion or preferably comparisons with other certified defenses need to be added.


Review 2

Summary and Contributions: The main contribution of this paper is to provide an algorithm to make a pre-tained model, which can be part of an API in a cloud, robust certifiably. The main promise is mentioned to be avoiding the re-training of the original classifier. The general idea is to use the Cohen's randomized smoothing as the basic building block and apply an adaptive denoising module right before feeding the input to the pre-tained classifier. The denoising module could be trained by the MSE loss, or CE loss of the model prediction of gaussian noise corrupted input and prediction of the original input. The latter is called STAB by the authors. The idea is applicable in both white-box and blackbox setting of the base classifier. In the blackbox setting, it is suggested to train a surrogate classifier for the STAB loss. The experimental results showed non-trivial performance (certified l2 radius) compared to the case where no denoising is applied prior to feeding the data to the network.

Strengths: The paper is written fairly well and is clear enough. The idea sounds simple and makes it possible to provide certified robustness of the pre-trained model.

Weaknesses: - By training a classifier on the original data in the white-box setting, the authors break the promise of avoiding any re-training that is given in the paper. - The main aim of this work is to be of practical use in the cloud services. I think certified robustness is generally believed to provide weaker robust accuracies than the Madry's adversarial training. This questions the applicability of this approach. - Limited novelty: The idea of using the input transformation in adversarial robustness is not novel (Lecuyer et al. (2018) as mentioned in the paper uses a similar idea). The only difference is that the authors are using this idea in the context of certified robustness and also making the transformation adaptive to the original classifier.

Correctness: Yes.

Clarity: Yes, the paper is written clearly.

Relation to Prior Work: Yes.

Reproducibility: Yes

Additional Feedback: I believe the authors should motivate their setting a lot better. They emphasize the need to avoid re-training and still do this in the blackbox setting! They also mention lp robustness, in which the certified radius is known to depend intrinsically on the input dimension, except for the l1 and l2 norms (e.g. https://arxiv.org/abs/2002.03517, and https://arxiv.org/abs/2002.08118). This further questions the practicality of this approach. Looking at the Fig. 3 and 4, it seems like the difference between the proposed method and the original Cohen's method, as an upper limit and ideal behavior, increases when the model becomes more complex (i.e. going from ResNet-18 to ResNet-50). Is there any explanation for this? This suggests that the method might not be scalable enough. In addition, the authors should give some insights on why the STAB+MSE loss gives more corrupted output compared to the MSE loss (Fig. 5), while the latter gives worse robustness radius.


Review 3

Summary and Contributions: The paper proposes denoised smoothing which prepends a customized denoiser before any pretrained classifier and then applies randomized smoothing to provably robustify the classifier. Particularly, the denoiser is trained by combining MSE and stability Objectives. Experimental results demonstrate that the proposed strategy can improve the provably robustness of the pretrained classifier under both white-box and black-box settings.

Strengths: - The goal to provably robustify arbitrary pretrained classifier without retraining the underlying weights is very interesting. The authors well motivate the practical usefulness of the new technique and support it with extensive experimental results. - The authors successfully demonstrate the effectiveness of denoised smoothing on public vision APIs like Azure, Google Cloud Vision, AWS, and Clarifai.

Weaknesses: - The proposed denoising method is quite straigthforward and thus the novelty of the technique itself is limited. Specifically, MSE objective is widely used in image denoising. Also, Stability Objective is used in Li et al. (2019) for improving the model robustness against large Gaussian noise, though the context is slightly different there. - There is still a small but clear gap between the new technique and randomized smoothing (Cohen et al. (2019)) even under white-box setting. However, it is not a big concern considering this is the first paper aiming for provably robustifying pretrained classifiers.

Correctness: Correct.

Clarity: Well written.

Relation to Prior Work: Clearly discussed.

Reproducibility: Yes

Additional Feedback: *** Post Rebuttal *** Thanks for the response to settle the questions and concerns from the other reviewers. I will keep my original score.

[Author Response · NeurIPS 2020]

We thank the reviewers for their comments. We address individual concerns below. If you think we address your
concerns, please consider raising the score.

**Reviewer 1:** *Simple method and limited contribution.* We believe the simplicity of our method is a strength, not a
weakness. In addition, it's not enough to just "plug and play" Cohen et al.'s method, as our No-Denoiser baseline shows
(Tables 1&2 and Figure 2). Thus our algorithmic contributions, though simple, are important.

*Relationship with prior work.* All prior work which apply a preprocessing step are empirical defenses that hope to
remove malicious perturbations by doing a preprocessing step. In our work, we apply a denoiser not to remove the
malicious noise, but to make the pre-trained classifier accurate under Gaussian perturbation of its input, therefore
making randomized smoothing effective when applied to this pre-trained classifier.

*The white-box scenario is misleading.* We believe Reviewer 1 misunderstood the meaning and context of "white-box"
in our paper. Whether the attacker has access to the denoiser or not doesn't affect our method as we aren't empirically
removing adversarial noise. Our guarantees are provable due to the use of randomized smoothing.

*Practicality: Performance gap w.r.t to Cohen et al's method.* One should expect our method, without retraining the
classifier, performs at most as well as Cohen et al. which trains the classifiers with smoothing in the loop. While the
gap between these methods point to important future work direction, the value of our contribution is clear by comparing
against the No-Denoiser baseline, which is the *real "plug-and-play"* referred to by the reviewer in point 1. See Tables
1&2 and Figures 2,3,&4.

*What threat models can the algorithm handle?* Since our method uses randomized smoothing, it can in theory handle all
threat models that randomized smoothing can handle (including l1, l2, linf, and Wasserstein). The only change in our
method would be the way our denoisers are trained. For instance, instead of training a denoiser to remove Gaussian
noise (for L2 certification), the denoiser shall be trained to remove noise sampled from other distributions (e.g. Laplace
distribution for L1 threat models). We agree this might be confusing in our paper so we will make sure to clarify it.

*No proper theoretical justification.* We are not sure what theoretical justification the reviewer is pointing to here. Our
method applies randomized smoothing to a composition of a classifier and denoiser, instead of only applying it to
a classifier as all prior works on randomized smoothing do. Therefore, all the theoretical guarantees of randomized
smoothing hold for us.

*The paper is tough to follow and read.* We will reorganize the paper to be more self-contained in the main text.

*Reproducibility* We provide detailed experimental details in Appendix B, along with a detailed code + pre-trained
models replicating all the experiments. So we are confused why the reviewer thinks the paper is not reproducible.

**Reviewer 2:** *The authors break the promise of avoiding any re-training that is given in the paper.* We stress that we
never re-train the base classifier neither in the white-box nor black-box settings. In the white-box setting, we assume
we know the base classifier, and we backpropagate through it, but we only update the denoiser. The whole purpose
of our paper is to get non-trivial certification results without re-training the classifier itself. We hope this clarifies the
confusion; we will update the paper accordingly.

*Comparison to Madry's adversarial training.* Madry's defense is empirical, whereas we are interested in certified
defenses. But in any case, Madry's defense requires adversarially training the classifier, whereas in our setting, the
classifier isn't allowed to be re-trained/modified at all. It is interesting to study whether PGD-like defenses can be
applied to pre-trained classifiers without modifying the latter, but this is outside the scope of this paper.

*Limited novelty.* We agree that our approach looks similar to the use of denoising auto-encoder from Lecuyer et al.
However, our approach is distinguished from Lecuyer et al in several ways:
1. The use of denoising auto-encoder in Lecuyer et al is solely aimed at speeding up training for certifying large models.
In our case, our motivation is to effectively apply randomized smoothing to pretrained models.
2. More importantly, we comprehensively investigate various training strategies (MSE/stability/classification objectives)
and application settings (white-box/black-box), which are not investigated in Lecuyer et al.

*Practicality issues.* Our approach utilizes randomized smoothing, thus any practicality issue of randomized smoothing
passes on to our approach. We agree with the reviewer on this point and we will make it clearer in the next revision.

*Gap between ours and Cohen et al. increases as the model becomes more complex.* This is indeed an interesting
observation that needs further investigation. It might be the case that larger architectures require different denoiser
architectures/training schemes. We don't claim we train our denoisers in the best way possible, and we believe with
improved training of the denoiser, the gap can be further reduced.

*Reconstruction artifacts of STAB+MSE compared to MSE* We think the reason for this is that STAB+MSE makes the
denoiser more customized to the base classifier, resulting in more corrupted reconstruction. MSE loss only considers
removing Gaussian noise, thus leading to visually better output. This requires further investigation and is left for future
work.

**Reviewer 3:** Thanks for your comments!

[Meta-Review · NeurIPS 2020]

The paper elicited a significant amount of discussion. Reviewers raised two key concerns about the paper: (1) The assumption of a specific kind of norm bound is overly limiting, and (2) The discussion of related work could be substantially improved. That said, the metareviewer and some of the reviewers appreciate the simplicity of the work, the fact that it is the first provable defense that doesn't require retraining, and the potential for future work based on the ideas here. For these reasons, we recommend acceptance. We strongly urge the authors to incorporate the feedback provided in the reviews. In particular, the authors should take particular care to clarify the increment over prior work on certified defenses and the references that the reviewers cite.